# THE GEOMETRY OF SPECTRAL GRADIENT DESCENT: LAYERWISE CRITERIA FOR SIGNSGD VS SPECSGD

**Hiroki Naganuma**[1,2]*, **Laura Gomezjurado**[3]*, **Mahdi Ghaznavi**[1,2]*, **Ioannis Mitliagkas,**[1,2]
[1]Mila, [2]Université de Montréal, [3]Stanford University,

## ABSTRACT

Optimization in deep learning has expanded beyond Euclidean methods to include entrywise sign updates (SignSGD) and spectral sign updates (SpecGD/Muon). While both can be viewed as steepest descent under non-Euclidean geometries ($l_\infty$ and spectral norms, respectively), existing theory analyzes them in isolation. This leaves a practical gap: for a specific layer with finite batch size, which geometry yields better optimization progress? We address this by deriving a unified bound on the expected one-step loss improvement that accounts for both the local geometry of the gradient signal and the geometry of the stochastic noise. We introduce the *Spec-Sign Advantage Index*, a layerwise scalar criterion that determines the optimal optimizer choice. Our analysis reveals that SpecGD is preferred not simply when gradients are low-rank, but when the nuclear-norm signal-to-noise ratio exceeds the $l_1$-norm signal-to-noise ratio, generalizing recent deterministic condition checks to the stochastic regime.

## 1 INTRODUCTION

The geometry of the optimization update plays a crucial role in training large-scale neural networks. While Euclidean Stochastic Gradient Descent (SGD) remains a baseline, optimizers operating in non-Euclidean geometries have gained prominence. **SignSGD** (Bernstein et al., 2018) utilizes element-wise signs, corresponding to steepest descent in the $l_\infty$ norm, and has shown robustness in distributed settings. More recently, **Spectral Gradient Descent (SpecGD)** and its momentum variant **Muon** (Jordan et al., 2024; Davis & Drusvyatskiy, 2025) utilize the polar factor of the weight gradient, corresponding to steepest descent in the spectral (operator) norm.

Despite their shared nature as "sign-based" methods—one entrywise, one spectral—theoretical analysis has largely treated them separately. Davis & Drusvyatskiy (2025) provide a condition comparing the gradient's nuclear rank to the activation's stable rank, but their analysis focuses primarily on the deterministic regime. Conversely, adaptive batch size analyses for non-Euclidean methods (Naganuma et al., 2026) highlight the critical role of gradient noise geometry but do not explicitly compare the relative advantage of one geometry over the other for a fixed computational budget.

**Contribution.** In this work, we propose a unified framework to compare SignSGD and SpecGD. We derive a noise-aware lower bound on the expected one-step loss improvement for both methods (Section 3). This allows us to define the **Spec-Sign Advantage Index** (Section 4), a computable ratio that indicates which optimizer is locally preferable. We show that the preference depends on a trade-off between the sparsity of the gradient signal and the structure of the stochastic noise, offering a granular, layer-wise heuristic for optimizer selection.

## 2 UNIFIED FRAMEWORK: GENERALIZED STOCHASTIC STEEPEST DESCENT

We consider the minimization of a loss function $L(W) = \mathbb{E}_\xi[l(W; \xi)]$ for a matrix parameter $W \in \mathbb{R}^{n \times m}$. Let $G = \nabla L(W)$ be the true gradient and $\widehat{G}$ be the stochastic mini-batch gradient computed on a batch of size $B$, such that $\mathbb{E}[\widehat{G}] = G$. We denote the noise matrix as $E = \widehat{G} - G$.

---

*Equal contribution. Authors are listed in alphabetical order.

Both SignSGD and SpecGD can be unified under the framework of **Stochastic Generalized Steepest Descent**. For a chosen primal norm $\| \cdot \|$, the update direction $D$ is chosen to maximize the alignment with the stochastic gradient under a unit dual-norm constraint:

$$D = \arg\max_P \langle \widehat{G}, P \rangle \quad \text{s.t.} \quad \|P\| \leq 1. \tag{1}$$

The update rule is $W_{t+1} = W_t - \eta D$. Table 1 summarizes the geometry for both methods. Note that $\| \cdot \|_*$ denotes the dual norm of $\| \cdot \|$.

Table 1: Geometric interpretation of SignSGD vs. SpecGD.

| Method | Update Direction ($D$) | Primal Norm ($\| \cdot \|$) | Dual Norm ($\| \cdot \|_*$) |
|--------|------------------------|------------------------------|------------------------------|
| SignSGD | $\text{sign}(\widehat{G})$ | $l_\infty$ (max entry) | $l_1$ (sum of abs) |
| SpecGD | $\text{polar}(\widehat{G})$ | Spectral ($\sigma_{\max}$) | Nuclear ($\sum \sigma_i$) |

## 2.1 ONE-STEP EXPECTED IMPROVEMENT

To compare these methods fairly, we look at the expected reduction in loss. Using a quadratic approximation of the loss $L$ with Hessian $H$, the expected improvement $\Delta(\eta) := \mathbb{E}[L(W) - L(W - \eta D)]$ is:

$$\Delta(\eta) \approx \eta \underbrace{\mathbb{E}[\langle G, D \rangle]}_{\text{Signal Alignment}} - \frac{\eta^2}{2} \underbrace{\mathbb{E}[\langle D, HD \rangle]}_{\text{Curvature Cost}}. \tag{2}$$

Comparing optimizers reduces to comparing the optimal improvement $\Delta^* = \sup_\eta \Delta(\eta)$. This requires bounding the Signal Alignment (how well the stochastic update captures the true gradient geometry) and the Curvature Cost (how much the loss penalizes that direction).

## 3 NOISE-AWARE SIGNAL ANALYSIS

The core difficulty in analyzing sign-based methods is that the update $D$ is a non-linear function of the noisy gradient $\widehat{G} = G + E$. We derive lower bounds for the signal alignment term $\mathbb{E}[\langle G, D \rangle]$ that explicitly account for the batch size $B$ and the noise geometry.

### 3.1 DEFINING NOISE SCALES ($c_1$ AND $c_*$)

We define the noise constants based on the aggregate variance of the gradient, following the non-Euclidean Gradient Noise Scale (GNS) theory (Naganuma et al., 2026).

**Definition 1** (Geometry-Specific Noise Scales). *Let $\Sigma$ be the covariance tensor of the gradient noise per sample. We define the noise constants $c_1$ and $c_*$ as the expected dual norm of the estimation error scaled by $\sqrt{B}$:*

$$c_1 := \sqrt{B} \cdot \mathbb{E}\left[\|\widehat{G} - G\|_1\right] \approx \|\sigma\|_1, \tag{3}$$

$$c_* := \sqrt{B} \cdot \mathbb{E}\left[\|\widehat{G} - G\|_{nuc}\right] \approx \text{tr}\left((\mathbb{E}[EE^\top])^{1/2}\right), \tag{4}$$

*where $\sigma$ is the vector of per-coordinate standard deviations, and the approximation for $c_*$ assumes row-wise covariance structure dominates (Naganuma et al., 2026).*

These constants quantify how "expensive" the noise is within the respective geometries. $c_1$ penalizes noise in every coordinate equally. $c_*$ penalizes noise aligned with the principal singular vectors.

### 3.2 LOWER BOUNDS ON SIGNAL ALIGNMENT

We now lower bound the expected alignment $\mathbb{E}[\langle G, D \rangle]$ for both optimizers.

**Proposition 1 (SignSGD Signal).** *For SignSGD, the expected alignment satisfies:*

$$S_{\text{sign}}(B) := \mathbb{E}[\langle G, \text{sign}(\widehat{G})\rangle] \geq \|G\|_1 - \frac{2c_1}{\sqrt{B}}. \tag{5}$$

*Proof Sketch.* Similar to the spectral case, we decompose $\langle G, \text{sign}(\widehat{G})\rangle = \|\widehat{G}\|_1 - \langle E, \text{sign}(\widehat{G})\rangle$. Applying the reverse triangle inequality $\|\widehat{G}\|_1 \geq \|G\|_1 - \|E\|_1$ and Hölder's inequality yields the bound with a factor of 2.

**Proposition 2 (SpecGD Signal).** *For SpecGD, the expected alignment satisfies:*

$$S_{\text{spec}}(B) := \mathbb{E}[\langle G, \text{polar}(\widehat{G})\rangle] \geq \|G\|_{\text{nuc}} - \frac{2c_*}{\sqrt{B}}. \tag{6}$$

*Proof Sketch.* Using the duality of norms, $\langle X, \text{polar}(X)\rangle = \|X\|_{\text{nuc}}$. We write $\mathbb{E}[\langle G, \text{polar}(\widehat{G})\rangle] = \mathbb{E}[\langle \widehat{G} - E, \text{polar}(\widehat{G})\rangle]$. Using the reverse triangle inequality for nuclear norms and Hölder's inequality on the noise term $\langle E, \text{polar}(\widehat{G})\rangle \leq \|E\|_{\text{nuc}}\|\text{polar}(\widehat{G})\|_{\text{op}}$, we obtain the bound dependent on $\mathbb{E}\|E\|_{\text{nuc}}$, which is $c_*/\sqrt{B}$.

## 4 THE SPEC-SIGN ADVANTAGE INDEX

We compare the optimal improvements. Let $H_{\text{sign}}$ and $H_{\text{spec}}$ be the expected directional curvature along the SignSGD and SpecGD paths, respectively:

$$H_{\text{sign}} = \mathbb{E}[\langle \text{sign}(\widehat{G}), H\,\text{sign}(\widehat{G})\rangle], \quad H_{\text{spec}} = \mathbb{E}[\langle \text{polar}(\widehat{G}), H\,\text{polar}(\widehat{G})\rangle].$$

Maximizing Eq. 2 with respect to $\eta$ yields $\Delta^* \approx \frac{S(B)^2}{2H}$. We define the layerwise advantage index $\mathcal{I}$ as the ratio of these optimal improvements.

**Definition 2** (Spec-Sign Advantage Index)**.** *For a layer with true gradient $G$, batch size $B$, and noise constants $c_1, c_*$, the advantage of SpecGD over SignSGD is given by:*

$$\mathcal{I}(G, B) := \frac{\Delta^*_{spec}}{\Delta^*_{sign}} \approx \underbrace{\left(\frac{\|G\|_{nuc} - 2c_*/\sqrt{B}}{\|G\|_1 - 2c_1/\sqrt{B}}\right)^2}_{\text{Signal Efficiency Ratio}} \times \underbrace{\left(\frac{H_{sign}}{H_{spec}}\right)}_{\text{Curvature Ratio}}. \tag{7}$$

*Decision Rule: If $\mathcal{I}(G, B) > 1$, SpecGD is preferred. If $\mathcal{I}(G, B) < 1$, SignSGD is preferred.*

### 4.1 ANALYSIS OF REGIMES

The Index $\mathcal{I}$ reveals three distinct regimes governing the choice of optimizer.

**1. The High-SNR / Deterministic Limit ($B \to \infty$).** As $B \to \infty$, noise terms vanish. The signal ratio becomes $(\|G\|_{\text{nuc}}/\|G\|_1)^2$. Since $\|G\|_{\text{nuc}} \leq \|G\|_1$ (with equality only for rank-1 diagonal matrices), SpecGD is at an inherent disadvantage regarding raw signal magnitude in coordinates. To win, SpecGD requires $H_{\text{spec}} \ll H_{\text{sign}}$. This occurs when the Hessian is aligned such that the sparse directions (favored by SignSGD) have high curvature, while the spectral directions have low curvature. This recovers the intuition from Davis & Drusvyatskiy (2025) regarding the condition $nr(G) \geq st(A)$.

**2. The Noise-Dominated Regime.** When $B$ is small, the noise terms $c_1$ and $c_*$ dominate. SpecGD typically has a structural advantage here. For a matrix of size $N \times N$, the $l_1$ noise $c_1$ sums variance over $N^2$ entries. The nuclear norm noise $c_*$ essentially sums variance over $N$ singular values (assuming noise is isotropic). Specifically, for Gaussian noise, $c_1 \propto N^2$ while $c_* \propto N^{1.5}$. **Implication:** Even if the true gradient is sparse (favoring SignSGD deterministically), SpecGD may be preferable at small batch sizes because the "spectral noise" accumulates slower than "coordinate noise."

**3. The Layer Shape Factor.** Consider a weight matrix $W \in \mathbb{R}^{n \times m}$. If $n \gg m$ (e.g., embedding layers), the matrix is naturally low-rank. Here, $\|G\|_{\text{nuc}} \approx \|G\|_F$, while $\|G\|_1$ can be up to $\sqrt{nm}\|G\|_F$. However, the noise $c_*$ is also much lower. The Index $\mathcal{I}$ provides a concrete way to balance these trade-offs without manual tuning.

## 5   DISCUSSION AND CONCLUSION

We presented a theoretical framework to compare entrywise SignSGD and spectral SpecGD/Muon updates. By analyzing the steepest descent objective under the lens of dual norms, we derived the *Spec-Sign Advantage Index*.

This index refines previous heuristics by explicitly incorporating the batch size $B$. It explains why spectral methods often outperform sign methods in the early phase of training (where gradients are large but noise is high, and the spectral filtering of noise via $c_*$ is beneficial) and why sign methods remain competitive for embedding layers (where coordinate sparsity is high).

While Davis & Drusvyatskiy (2025) proposed the condition $nr(G) \geq st(A)$ (Nuclear Rank $\geq$ Stable Rank) for spectral updates, their derivation largely assumes exact gradients. Our term $(\|G\|_{\mathrm{nuc}} - \mathrm{noise})^2/(\|G\|_1 - \mathrm{noise})^2$ can be seen as a stochastic generalization of their ratio. Specifically, when noise is high, the "effective" nuclear rank of the gradient degrades slower than the "effective" $l_1$ geometry, expanding the regime where spectral updates are advantageous.

**Limitations.** Similar to Davis & Drusvyatskiy (2025); Balles et al. (2020), our analysis relies on local quadratic approximations of the loss landscape, which may not fully capture the complex, non-convex dynamics of deep neural networks over long trajectories. Additionally, the computational cost of evaluating the Advantage Index—specifically the requirement for singular value decompositions ($c_*$) and Hessian projections ($H_{\mathrm{spec}}$)—may be prohibitive for real-time adaptive switching in its current form. Most importantly, the findings presented here are theoretical. Extensive numerical experiments are required to quantify the wall-clock performance gains of switching strategies in practice.

Future work involves integrating this index into an adaptive optimizer that switches geometries per-layer dynamically during training, as suggested by the varying noise-to-signal ratios observed in training transformers.

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
