# OpenReview forum: "The Geometry of Spectral Gradient Descent: Layerwise Criteria for SignSGD vs SpecSGD"
_ICLR.cc/2026/Workshop/GRaM — ICLR 2026 Workshop GRaM Poster_

### Official Review · Reviewer_rH7v · 2026-02-20

**Rating:** 6
**Confidence:** 5

**Review:**

Objective: This paper tries to build a unified theoretical framework comparing SignSGD and Spectral GD (e.g. Muon) with geometric specific noise scales; coordinate noise ($c_1$) and nuclear noise ($c_*$) and introducec Spec-Sign Advantage Index, which can act as an indicator as to when SpecGD is beneficial over SignGD.

PROS:
1. Lower Bounds Signal Alignment for SignGD and SpecGD.
2. Proposes an Indicator Function (Definition 2) to choose between SignGD and SpecGD.

CONS:
1. Is there any relationship between the signal alignment of SignGD and SpecGD?
2. Can you at least give one experiment validation (synthetic or MNIST) (let's say the data is Gaussian random with high variance) and show how SignGD, SpecGD and the proposed Indicator Function help in optimization speedups.

Relevance to topics listed in GRaM call for papers: Yes

Originality and novelty: Yes

Technical soundness of method: Yes

Clarity in writing and organization: Good, up to the mark!

For the Proceedings track: N/A

Double-blind reviewing: No violations of anonymity were found.

Use of LLMs: The text is technical and precise; there are no signs of excessive or improper LLM generation.

**Pmlr Suitability:**

NA

---

### Official Review · Reviewer_2BVE · 2026-02-24
**The Geometry of Spectral Gradient Descent: Layerwise Criteria for SignSGD vs SpecSGD**

**Rating:** 6
**Confidence:** 3

**Review:**

This paper develops a theoretical framework to compare entrywise sign updates with spectral sign updates. The authors derive lower bounds on expected one-step improvement that separate signal alignment and curvature cost, with explicit dependence on batch size via geometry-specific noise. They then introduce a layerwise criterion, the Spec–Sign Advantage Index, that determines the optimal optimizer choice.


### Strengths
- The paper provides a unifying viewpoint by casting both SignSGD and SpecGD as stochastic steepest descent under dual norms, which clarifies what differs and what is shared.
- The proposed Advantage Index is interpretable and captures the finite-batch noise, which is an important factor missing from many deterministic comparisons.
- The analysis of regimes yields actionable qualitative predictions that align with common practitioner intuitions about early training and layer types.

### Weaknesses
- The analysis is entirely local (quadratic approximation) and one-step, so it does not justify long-horizon behavior in nonconvex training, where geometry and noise evolve.
- Without experiments, it is unclear whether the Spec–Sign Advantage Index predicts wall-clock gains or robust optimizer switching decisions in practice.

**Pmlr Suitability:**

NA

---

### Meta-Review · Area_Chair_GoQm · 2026-02-23

**Decision:**

Accept

**Metareview:**

The authors compare entrywise sign (SignSGD) and spectral sign (SpecGD/Muon) as non-euclidean optimizers, unify them via dual norms. They compare them as a function of batch size and gradient noise and introduce a Spec–Sign Advantage Index to choose which one is locally better. They further find that SpecGD tends to win in the small-batch and noisy regime when the nuclear-norm signal-to-noise decays less with noise than the $l_1$ signal-to-noise.

As noted by the reviewer, the work is **novel, meaningful, very well written**. It is a clear accept.

**Relevance To Proceedings:**

Tiny paper — does not apply

**Relevance To Workshop:**

Yes — suitable for GRaM

---

### Decision · Program_Chairs · 2026-03-02

Accept (Poster)